



# Heterogeneous nucleation of water vapor on different types of black carbon particles

Ari Laaksonen[1,2], Jussi Malila[3], Athanasios Nenes[4,5]

[1]Finnish Meteorological Institute, 00101 Helsinki, Finland.

[2]Department of Applied Physics, University of Eastern Finland, 70211 Kuopio, Finland.

[3]Nano and Molecular Systems Research Unit, 90014 University of Oulu, Finland.

[4]Laboratory of Atmospheric Processes and their Impacts, School of Architecture, Civil & Environmental Engineering, École
Polytechnique Federale de Lausanne, 1015, Lausanne, Switzerland

[5] Institute of Chemical Engineering Sciences, Foundation for Research and Technology Hellas (FORTH/ICE-HT), 26504,
Patras, Greece

*Correspondence to*: Ari Laaksonen (ari.laaksonen@fmi.fi)

**Abstract**: Heterogeneous nucleation of water vapor on insoluble particles affects cloud formation, precipitation, the

hydrological cycle and climate. Despite its importance, heterogeneous nucleation remains a poorly understood phenomenon

that relies heavily on empirical information for its quantitative description. Here, we examine heterogeneous nucleation of

water vapor on and cloud drop activation of different types of soots, both pure black carbon particles, and black carbon

particles mixed with secondary organic matter. We show that the recently developed adsorption nucleation theory

quantitatively predicts the nucleation of water and droplet formation upon particles of the various soot types. A surprising

consequence of this new understanding is that, with sufficient adsorption site density, soot particles can activate into cloud

droplets – even when completely lacking any soluble material.



## 1  Introduction

When water vapor becomes supersaturated – i.e., its relative humidity exceeds 100% – it is in a metastable state, and can

form liquid water or ice. External surfaces can facilitate the phase transition, through a process known as heterogeneous

nucleation. Industrial processes (e.g. dropwise condensation), biological systems (e.g. infection strategy of plant pathogens),

everyday situations (e.g. fogging of glasses and windshields), and fundamentally important atmospheric phenomena (clouds,

frost and hoar frost) are all driven by the heterogeneous nucleation of water (Pruppacher and Klett, 1997; Franks, 2003).

Despite its ubiquity and importance, heterogeneous nucleation of water vapor remains poorly understood even after more

than a century of research (Möller, 2008). This poor understanding is expressed by the lack of an established heterogeneous

nucleation theory that provides quantitative comprehension of the process, and translates to a large uncertainty regarding the

role of aerosol–cloud interactions in the climate system (Seinfeld et al., 2016). All traditional formulations are variants of

classical nucleation theory (CNT) (Fletcher, 1958), which provide notoriously poor predictions of water drop nucleation

(Mahata and Alofs, 1975). Molecular simulations (Lupi et al., 2014) are able to reveal aspects of heterogeneous nucleation

phenomena, but on their own cannot provide a theoretical framework for describing heterogeneous nucleation in atmospheric

and climate models – and are impractical for implementation in models themselves.

A shortcoming of CNT is that it does not recognize the existence of any water on surfaces prior to the formation of a

macroscopic droplet (Fig. 1a). Another shortcoming is that all the energetics of interaction between the surface and the

nucleating droplet are expressed by one parameter, the contact angle (Fletcher, 1958). In reality, water adsorbs to the surface

already at sub-saturated conditions (relative humidity below 100%), which can play a critical role in the nucleation process.

Notably, the energetics of adsorbed water differs considerably from what is implied by the single value provided by the

contact angle. In fact, the heat of adsorption changes with increasing adsorption layer thickness (Hill, 1949a). Not describing

the process of water adsorption therefore omits important physics that may hinder a quantitative description of

heterogeneous nucleation. Unlike CNT, the adsorption nucleation theory (ANT) (Laaksonen, 2015; Laaksonen and Malila

2016) accounts for water adsorption prior to onset of heterogeneous nucleation (Figs. 1a and 1b).



Soot particles formed in combustion processes often contain sufficient amounts of water-soluble impurities so that their CCN activity can be described using the kappa-Köhler theory (Petters and Kreidenweis, 2007), provided that the kappa-value is

known. An exception is the major fraction of soot particles produced by aircraft engines, which contain only small amounts of impurities, and are rather hydrophobic (Demirdjian et al., 2006). As those particles may participate in contrail or cirrus ice formation (Ikhenaze et al, 2020) either directly after emission or at some later point in time, it is important to be able to characterize their water uptake and nucleation properties theoretically ad well as possible.

Water insoluble atmospheric particles often have rough surfaces characterized by cavities that display physical and chemical

heterogeneity. This means that nucleation is not equally likely for all locations on the surface, which is another challenge for CNT. Here we show that when both adsorption and appropriate geometric considerations are included in the theoretical description of water nucleation, a quantitative theory of heterogeneous nucleation of water on aerosol is established. We demonstrate the new framework for different types of black carbon (BC) particles, which are long known to be climate active – but with a large uncertainty on the cumulative impact.






a)                                              b)

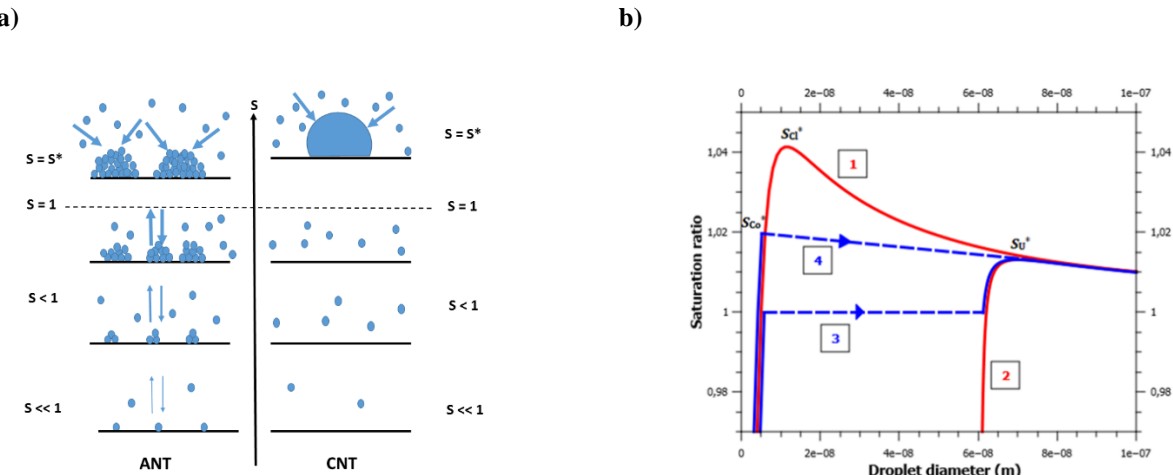

**Figure 1. a) A schematic showing the principal differences between ANT and CNT. ANT is based on the premise that**

**vapor molecules are adsorbed on the surface already at subsaturation, and the molecular nature of adsorption is**

**reflected in the values of the FHH parameters $A$ and $B$ that are obtained from experimental adsorption equilibria. The**

**premise of CNT is that liquid droplets appear on the surface once the vapor becomes supersaturated. When**

**experimental contact angles are used. CNT predicts higher critical supersaturations S\*, at which unlimited**

**condensation starts, than ANT. In practice, the contact angle used for CNT is an adjustable parameter to fit**

**experiments, while for ANT it is the bulk (observed) value, adjusted owing to the interaction of adsorbed water**

**molecules on the surface. b) Schematic Köhler type curves for various heterogeneous nucleation pathways of water on**

**a 60 nm spherical particle. Curve 1 depicts the saturation ratio as a function of the diameter of individual droplets with**

**contact angle of 30°. Nucleation takes place at the maximum of the curve. Curve 2 is the Köhler curve for the FHH**

**activation mechanism, corresponding to a contact angle of 0°. Curve 3 corresponds to a situation where the droplets**

**fill the surface of the solid particle just below S = 1, and coalesce to form a uniform liquid film, which then grows along**

**curve 2 until nucleation occurs at the maximum. Curve 4 is otherwise similar to curve 3, but the droplets fill the surface**

**at S = 1.02 and nucleation occurs instantaneously because the saturation ratio is higher than the maximum of curve 2.**

**Note that the blue curves have been slightly offset along the x-axis for clarity.**



## 2 Theoretical framework

It has been shown with atomic force microscopy (Cao et al., 2011) that adsorbed water exists as discrete "patches" (droplets)

of liquid water, rather than a uniform film, on non-wettable surfaces. These patches then grow with increasing relative

humidity (RH). Unless they fill the surface and coalesce to form a film, the growth continues past 100% RH to the

supersaturated regime (Rowley and Innes, 1942), owing to the Kelvin effect, which causes the equilibrium vapor pressure to

increase over curved surfaces. At a characteristic level of supersaturation (the so called "critical supersaturation"), however,

the droplets experience heterogeneous nucleation, where unconstrained condensation of water vapor ensues to form a water

droplet.

To account for the above considerations, we develop a theory that is based on physisorption and captures the progressive

accumulation of water, from monolayer "patches" on a surface (at low RH) up to macroscopic amounts of condensed

material throughout the adsorption–nucleation transition (at the critical supersaturation). For this, we adopt a multilayer

formulation that has minimal coefficients and can be informed by standard adsorption experiments. Our formulation

(Laaksonen, 2015; Laaksonen and Malila, 2015; Laaksonen et al., 2016) uses a combination of the Frenkel–Halsey–Hill

(FHH) multilayer adsorption theory (Frenkel, 1946; Halsey, 1948; Hill, 1949b) and the Kelvin effect (Defay et al., 1966).

We treat the surface as covered with individual, adsorbed droplets with an average distance $s$ from their nearest neighbours.

If the average droplet size and the average distance between them is known, the experimentally observable adsorption layer

thickness can be calculated. Surface heterogeneities imply droplets with a distribution of radii and contact angles; but as a

first approximation we assume all droplets to have the same size and contact angle (these can be considered as average

values). The equilibrium condition for a single droplet is given by

$$\ln(S) = -\frac{A}{\overline{N_d}^B} + \frac{2\gamma v_w}{kTR} \tag{1}$$

where $S$ denotes saturation ratio, $\gamma$ is surface tension, $v_w$ is the volume of a water molecule, $k$ is the Boltzmann constant, $T$ is

temperature, $R$ is the droplet radius, and $A$ and $B$ are FHH parameters. The average number of monolayers, $\overline{N_d}$, over the

droplet projection area on the adsorbent surface is given by $\overline{N_d} = \overline{\delta}/\delta_m$, where $\overline{\delta}$ denotes the average water layer



thickness, and $\delta_m$ is the thickness of a monolayer given as $v_w/\sigma$, where $\sigma$ is the cross-sectional area of an adsorbed water

molecule. $\overline{N_d}$ and $R$ can be related via a relation that depends on the contact angle $\theta$ and the curvature of the adsorbent

surface (see Materials and Methods for details of the theory). Thus, unlike in CNT, contact angle is a purely geometric

parameter describing the behavior of bulk water on the surface. Molecular simulations have shown that the contact angle of a

water nanodroplet on graphitic surface equals the macroscopic value for droplets having radii larger than about 4 nm (Sergi

et al, 2012). For nanodroplets on hydrophilic clays, the contact angle does not vary for droplets containing 256 – 1000

molecules (Zheng et al., 2017) – which means that any change in contact angle refers to nanodroplets with radii of the order

of 1 nm. The sizes of critical clusters calculated using our theory are typically much larger than that, justifying the use of

experimental contact angles in the heterogeneous nucleation calculations.  In case of a zero contact angle, water forms a

uniform film on the adsorbent surface. If uniform adsorption occurs on spherical particles with radius $R_p$, the average number

of monolayers is $\overline{N}$ = $(R-R_p)/\delta_m$ (Sorjamaa and Laaksonen, 2007; Laaksonen and Malila, 2016).

The FHH interaction parameters $A$ and $B$ describe the strength of molecular interaction between the adsorbent and the first

adsorption layer, and the exponential decay of the interaction strength as a function of distance, respectively. With the van

der Waals model fluid, $A$ and $B$ have well defined values (note, however, that $A$ depends on the assumed molecular cross-

section of the adsorbing molecule), but in real systems they must be treated as empirical constants that can be determined

from measured adsorption isotherms. In case of droplet-wise adsorption, the macroscopically determined adsorption layer

thickness is a statistical measure that can be related to the droplet size if the average distance between the adsorbed droplets,

$s$, and the contact angle are known.  In practice, $s$ can be obtained together with $A$ and $B$ from the adsorption isotherms

(Laaksonen, 2015; Laaksonen and Malila, 2016).

We note that there is considerable uncertainty with the value of the experimental contact angle, due to contact angle

hysteresis (Tadmor, 2004). The difference between the advancing and receding angles can be on the order of 10 degrees or

even more. In any case, with graphite the critical supersaturation is surprisingly insensitive to an uncertainty of around $\pm\ 5^\circ$

in the value of $\theta$. The reason for this is that the contact angle is included in the equations used when the theory is fitted to

adsorption experiments in order to determine the values of the FHH parameters $A$ and $B$. Thus, if the contact angle is



changed by a few degrees, the model has to be re-fitted to the adsorption data, which can be done simply by changing the value of the $A$-parameter. It turns out that this does not either compromise the quality of the fit to the adsorption experiments or noticeably change the calculated critical supersaturation unless the contact angle is changed by more than about plus or minus five degrees (see Fig. A2).

Figure 1b shows a schematic picture of nucleation of water vapor on a spherical particle in our model framework. A droplet grows along the red curve 1 until it reaches the critical supersaturation $S_{cl}^*$ and nucleates. However, if the average distance between the droplets is small enough, and they coalesce to form a uniform liquid film at sufficiently low saturation ratio (blue curve 3), the new equilibrium as well as the new critical supersaturation $S_U^*$ can be calculated simply by using a zero contact angle (red curve 2). Depending on the value of $s$, it is also possible that the droplet coalescence occurs at some

supersaturation $S_{co}^*$ between $S_U^*$ and $S_{cl}^*$, in which case nucleation is immediate (blue curve 4).

## 2.1 Adsorption nucleation theory

The starting point of the theory is that adsorbing vapor molecules are assumed to cluster around adsorption sites to form spherical caps of liquid drops having a contact angle $\theta$. The equilibrium condition is given by Eq. (1). The geometry for a

spherical seed particle is as shown in Fig. 2. The average over $\delta^B$ is given by

$$\overline{\delta^B} = \left[1 - \cos\Phi\right]^{-1} \int_0^\Phi \delta^B \sin\alpha \, d\alpha \qquad (2)$$

with

$$\delta = -R_p + R\cos\beta + \sqrt{(R\cos\beta)^2 - R^2 + d^2} \qquad (3)$$

$$d = \sqrt{R_p^2 + R^2 - 2R_p R\cos\theta} \qquad (4)$$

$$\cos\Phi = (R_p - R\cos\theta)/d \qquad (5)$$

$$\cos\beta = (R_p + \delta - d\cos\alpha)/R \qquad (6)$$





$S$ can now be computed as a function of $R$ using Eq. (1) with $\overline{N_d^B} = \overline{\delta^B}/\delta_m^B$ evaluated numerically from Eqs. (2) – (6). The resulting curve shows a maximum at the critical supersaturation $S^*$ (and critical radius $R^*$), marking the onset of heterogeneous nucleation.


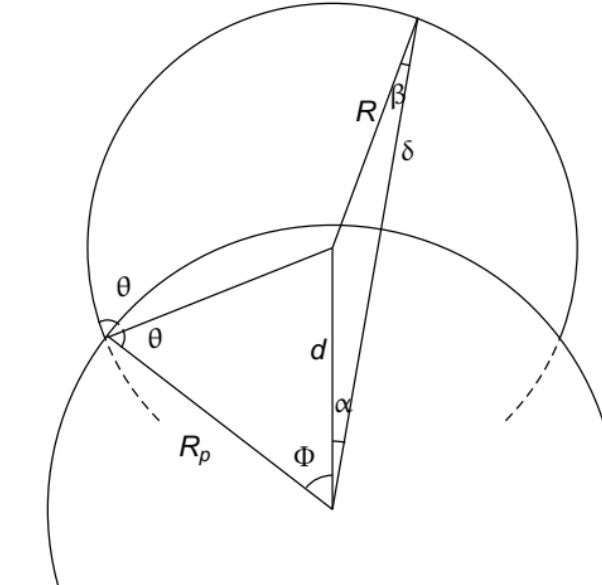

**Figure 2. The geometry used in deriving equations for nucleation of a spherical drop on a spherical nanoparticle.**

The values of the adsorption parameters $A$ and $B$, as well as that of the distance between active sites $s$, can be obtained from

measured adsorption isotherms. If a portion of the adsorption data is clearly in the multilayer regime and the data points align linearly when $\ln(-\ln(S))$ is plotted vs. $\ln(N)$, parameters $A$ and $B$ can be determined by fitting the classical FHH equation $\ln(S) = -AN^B$ to the data (here, $N$ denotes the macroscopically observable surface coverage). If the contact angle is known, the distance between active sites can be obtained from the sub-monolayer part of the adsorption isotherm. To do that, a relation is



needed between the size of an individual adsorbed droplet, and the macroscopic surface coverage. Making the approximation

$\overline{N_d^B} \approx \overline{N_d}^B$, Eq. (1) can be translated into

$$\ln(S) = -A\left[\frac{\pi\zeta^2}{\eta^2 N}\right]^{B/3} + \frac{2\gamma}{3kT}\left[\frac{\pi\eta\zeta^2}{N}\right]^{1/3}, \qquad \theta \leq 90°, \qquad (7)$$

$$\ln(S) = -A\left(\frac{\eta}{\beta}\right)^B \left[\frac{\pi g(\theta)}{3s^2 N}\right]^{\frac{B}{3}} + \frac{2\gamma v}{3kT}\left[\frac{\pi g(\theta)}{3s^2 N}\right]^{\frac{1}{3}}, \quad \theta > 90° \qquad (8)$$

with $\zeta = 3v\sin\theta/s$, $\eta = \sigma f(\theta)$, $g(\theta) = 4 - (1 + \cos\theta)^2(2 - \cos\theta)$, and $f(\theta) = \frac{(1 - \cos\theta)^2(2 + \cos\theta)}{\sin^2\theta}$, $\theta \leq 90°$ ;

$f(\theta) = 2 - 3\cos\theta$, $\theta > 90°$.

Distance between adsorption sites $s$ can now be determined by fitting Eq. (7) or (8) to the sub-monolayer data (note that it is

best to use the portion of the data just below a monolayer, as the low coverage data may be affected by pores or other factors

not taken into account in the above equations.

When the adsorbent surface is poorly wettable, the adsorption data may not extend to the multilayer regime at all. In such a

case, all of the parameters need to be obtained from a best fit of either Eq. (7) or Eq. (8) to the data set.

As noted in relation to Fig. 1b, nucleation can also occur due to coalescence of adsorbed droplets. The criteria we use for the

coalescence nucleation are that (i) the projection areas of adsorbed droplets equal the surface area of the seed particle, and that

(ii) the supersaturation exceeds the critical supersaturation calculated for a seed particle that is perfectly wettable.

**2.2 Liquid drop nucleation on insoluble particles mixed with soluble material**

If $\theta = 0°$, i.e. the seed particle is completely wettable, the saturation ratio can be written in terms of dry and wet particle radii

as

$$\ln(S) = -A\left(\frac{R - R_p}{D_w}\right)^{-B} + \frac{2\gamma v_w}{kTR} \qquad (9)$$

where we have approximated monolayer thickness with the diameter of a water molecule, $D_w$ (Laaksonen and Malila, 2016).

For a partially soluble seed particle – for example, a black carbon particle coated with low-volatility organic or inorganic





substances after atmospheric processing – Eq. (9) needs to be further modified to take the hygroscopicity of solution into

account. The following equation (Kumar et al, 2011) can be written in terms of the hygroscopicity parameter $\kappa$ (Petters and

Kreidenweis, 2007):

$$\ln(S) = -A\left(\frac{R-R_p}{D_w}\right)^{-B} - \frac{\varepsilon_s R_p^3 \kappa}{(R^3 - \varepsilon_i R_p^3)} + \frac{2\gamma v_w}{kTR} \tag{10}$$

where $\varepsilon_s$ and $\varepsilon_i$ are the volume fractions of soluble and insoluble species in the seed particles.


### 3. Results and discussion

### 3.1 Constraining water adsorption parameters on carbon

We now focus on determination of water vapor adsorption parameters on graphite and on black carbon (BC) particles. Figure

2 shows a logarithmic FHH plot of several adsorption isotherms. Starting from the bottom, the lowermost measurement data

set (Young et al, 1954) shows adsorption on graphitized carbon black treated at 400 °C for 12 hours prior to the

measurements.  Based on the minute amounts of water adsorbed at even the highest relative humidity, and information

obtained from heat of immersion measurements, Young et al. (1954) suggested that clusters of water adsorb at hydrophilic

surface sites which are far apart and may arise from surface oxides formed at the edge atoms of graphite, which has since

then been corroborated by quantum chemical calculations (Oubal et al., 2010). This description is in complete agreement

with the droplet-wise FHH adsorption model (see the lowermost solid line in Fig. 3 where the model reproduces the data).

The model parameters are shown in Fig. 3; the contact angle value was obtained from Fowkes and Harkins (1940), and the

parameters $A$, $B$, and $s$ were determined by optimizing the model fit to the experimental data.





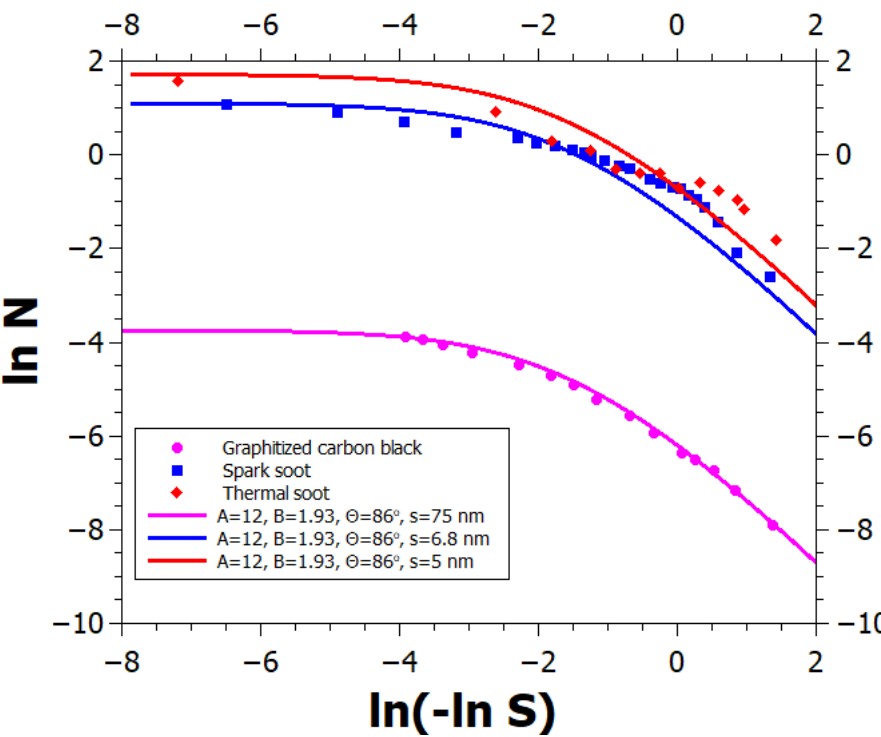

**Figure 3. An FHH-plot of experimental (markers) and modelled (lines) adsorption isotherms of water vapor on graphite and different soots. Note that in the abscissa, increasing supersaturation is to the left. See text for details.**

When BC is exposed to atmospheric conditions, the number of hydrophilic surface sites is expected to increase with time as different contaminants adsorb on the carbon and may undergo oxidation (Aria et al., 2016). Contaminants may also be present due to imperfect combustion, in case the carbon was produced by burning. This suggests that adsorption isotherms of BC containing impurities can be accounted for by increasing the density of the adsorption sites in the FHH model (without changing the other adsorption parameters), constrained by observations.

The data in the upper part of Fig. 3 (Kuznetsov et al., 2003; Popovicheva, 2008) show water adsorption on spark discharge soot, and thermal soot produced by natural gas pyrolysis. The materials were not treated at high temperature prior to the



adsorption measurements, and are thus likely to contain more adsorption sites from surface contaminants than the

graphitized carbon black used by Young et al. (1954). The blue and red solid curves represent the corresponding adsorption

isotherms, obtained by simply reducing the average distance of adsorption sites in the FHH model from 75 nm to 6.8 nm and

5 nm, respectively, and keeping the other parameters at the graphitized carbon black values.  The data for RH > 80% (ln(-ln

($S$)) < -1.5) agree well with the theoretical curves, whereas at lower relative humidities the data show more adsorption than

the model. The latter is caused by the effects of soot porosity, which is indicated by a hysteresis between the measured

adsorption and desorption curves.

When the density of hydrophilic adsorption sites is sufficiently high, the growing water clusters will start coalescing as the

RH increases above some threshold value, and a liquid film will be formed on the BC surface. If the film formation occurs at

RH below 100%, the conditions for activation of a BC particle into a cloud droplet (where the particle is said to act as a

Cloud Condensation Nuclei, or CCN) can be described using a zero contact angle and the total surface area of the particle,

while the full adsorption nucleation theory with non-zero contact angle is required when the active site density is low (curve

3 in Figure 1b). We next examine whether the CCN activation of different types of BC particles can be described with the

theoretical framework as constrained above, using model parameters shown in Fig. 3.

**3.2 Water nucleation on black carbon particles**

Figure 4 depicts CCN activation data for hydrophobic and hydrophilic BC particles. The most hydrophobic particles

(Kotzick et al., 1997) were produced using a spark discharger. The particles were in contact only with argon and nitrogen

gases prior to the activation measurements (black spheres), and thus contained very little adsorption sites. As can be seen, all

model curves are below the spark discharger data. However, the model with parameters obtained using the adsorption data

for graphitized carbon black (Fig. 3) is in excellent agreement with the CCN activation data for spark discharger particles

that were subjected to ozonolysis prior to the activation measurement (Kotzick et al., 1997) (red squares), and therefore have

an increased density of adsorption sites.

**Figure 4. Experimental (markers) and theoretical (lines) critical supersaturations as a function of particle size for cloud drop activation of hydrophobic and hydrophilic soot particles. See text for details.**

The blue diamonds in Fig. 4 show activation of soot particles produced by burning camp stove fuel (white gas) (Hagen et al., 1989). Such particles can be expected to contain a large amount of oxidized species on their surfaces, and may therefore already have a liquid film of water at activation. The blue curve shows model prediction with the same FHH parameters $A$ and $B$ as the red curve, but a zero contact angle. It appears that the model has somewhat gentler size dependence than the



rather scattered data, but otherwise the agreement is quite satisfactory. The CCN activation of pigment black particles (Dalirian et al., 2018), also match the theoretical line within uncertainty limits (with the exception of the largest particle size, which has the uncertainty range just below the theoretical line).

Very interestingly, the three datapoints of Dusek et al. (2006) are clearly above those of Dalirian et al. (2018). The principal
difference between the soots used in these two studies is that Dusek et al. (2018) applied heat treatment, thus very likely reducing the adsorption site density. Indeed, a theoretical line calculated with average adsorption site distance 2.8 nm and assuming the coagulation nucleation to occur, matches the data perfectly. When a site distance of just 0.2 nm smaller is assumed, the theoretical prediction becomes equal to the CCN activation prediction for particles larger than about 50 nm. With smaller particles, the coagulation nucleation appears to match the camp stove gas soot data better than the CCN
activation theory. In general, accounting for droplet-wise adsorption and resulting alternative pathways for heterogeneous nucleation (Fig. 1) allows us to obtain a much-improved description on the nucleation of water on BC particles than CNT (Kotzick et al., 1997) or adsorption nucleation theories not accounting for initial adsorption on distinct sites (Henson, 2007).

Further enhancement in BC hydrophilicity, on top of increased adsorption site density due to contaminant oxidation, may occur e.g. as a result of heterogeneous chemistry leading to a substantial amount of water-soluble material on the BC
particles. The red diamonds in Fig, 5  (Wittbom et al., 2014) are for particles aged in a smog chamber prior to the activation measurements, and were estimated to contain between 8 and 91 percent by mass of water soluble substance, respectively (the data points were deciphered from Figs. 10 and 11 in Wittbom et al. (2014)). The blue crosses were calculated using the theory which describes the combined effect of adsorption and water-soluble material when the volume fraction $\varepsilon$ and the hygroscopicity parameter $\kappa$ of the water-soluble material are known (Kumar et al., 2011; Petters and Kreidenweis, 2007; see
previous section for details). The FHH parameters were kept the same as for the pure black carbon particles, and the κ-value was assumed to be 0.13 (Wittbom et al., 2014).  The model predictions, shown by the blue crosses, are once again in excellent agreement with the data. Only in three cases the theory provides clear underestimates, the worst occurring at the lowest SOA mass fraction of about 0.08, as can be seen in Fig. 5b, while Fig. 5a shows that the experimental point is very close to the Kelvin line. Comparison to Fig. 4 indicates that this data point is compatible with activation of particles that do
not contain water soluble material.



a                  b

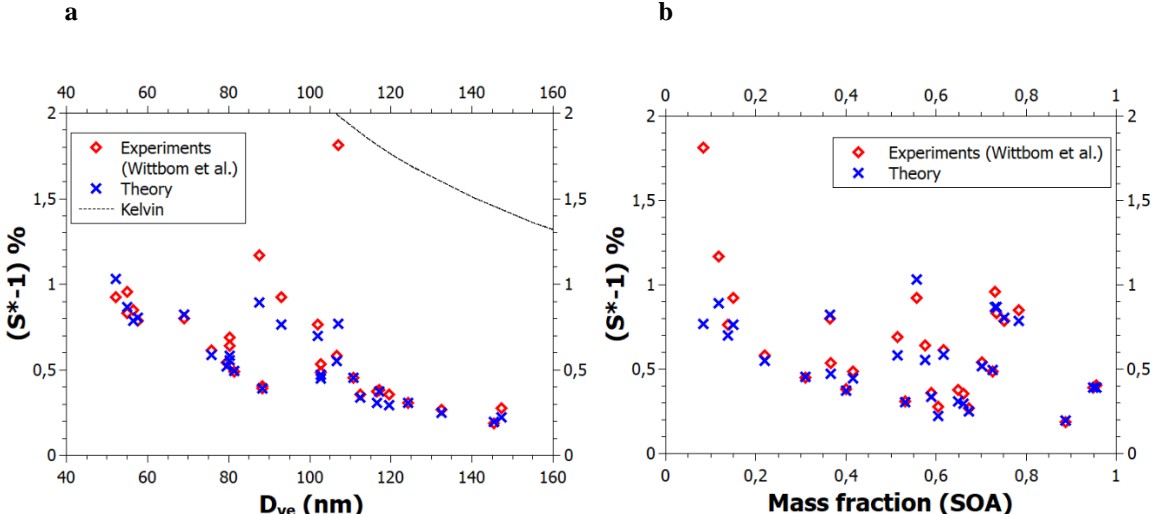

**Figure 5. Experimental and theoretical critical supersaturations of water vapor on mixed soot particles containing BC and SOA as a function of a) volume equivalent diameter of dry particles and b) SOA mass fraction. The data are a subset of the data shown in Figures 10 and 11 of Wittbom et al. (2014), and were identified by matching supersaturations (to a precision < 0.01%) in the two figures.**

## 4. Conclusions

In this study, we have shown that the adsorption nucleation theory can predict the critical supersaturations for heterogeneous nucleation and CCN activation of water vapor on different types of black carbon particles, using a single set of experimental adsorption and contact angle parameters.

It is quite obvious that surface heterogeneities cause adsorbed droplets to have a range of radii and contact angles, but to make the theory usable, we assume that we can resort to average values. This applies to the FHH parameters $A$ and $B$ as well; it is likely that there would be variability if isolated surface regions were examined. However, the variability of the heterogeneities – whether chemical or physical – is reflected in the values of the "average" adsorption parameters obtained from macroscopic adsorption isotherms. Our hypothesis is that the same parameters can be made use of when extending the theory to the supersaturated regime to predict critical supersaturations. The good match between our theoretical predictions





and the experimental critical supersaturations provides clear indication that our hypothesis is well supported. This also

implies that the heterogeneities occur at a spatial scale smaller than the surface area covered by the critical nuclei.

Our findings concerning adsorption nucleation on BC have a direct relevance for climate studies, as even the sign of the

radiative forcing due to BC aerosol–cloud interactions remains uncertain (Bond et al., 2013). Notably, atmospheric lifetime

of BC particles is regulated by condensation of water and subsequent cloud processes, which contributes further to the

uncertainty of the total radiative forcing (Samseth et al., 2018).

**Appendix A: Water properties used in the calculations**

$T$ denotes absolute temperature (K), $T'$ is temperature in degrees Celsius, and $R_g$ is the molar gas constant.

Equilibrium vapor pressure of liquid water (Pa) (Murphy and Koop, 2010):

$p_w$ = exp{54.842763 - 6763.22/$T$ - 4.21ln($T$) + 0.000367$T$ + tanh[0.0415($T$-218.8)][53.878-1331.22/$T$-9.44523ln($T$)+0.014025$T$]}

Density of liquid water (g cm$^{-3}$) (Marcolli, 2017):

$\rho_w$ =   1.8643535 − 0.0725821489$T$ + 2.5194368·10$^{-3}$$T^2$ − 4.9000203·10$^{-5}$$T^3$ + 5.860253·10$^{-7}$$T^4$ −

4.5055151 ·10$^{-9}$$T^5$ + 2.2616353·10$^{-11}$$T^6$ − 7.3484974·10$^{-14}$$T^7$ +1.4862784·10$^{-16}$$T^8$ −

1.6984748·10$^{-19}$$T^9$ + 8.3699379·10$^{-23}$$T^{10}$

Surface tension of water (N m$^{-1}$) (Hrubý et al., 2014):

$\gamma_w$ =   0.2358(1− $T$/647.096)$^{1.256}$[1− 0 .625(1− $T$/647.096)]

**Appendix B: Sensitivity analysis**

The contact angle value used in the calculations shown above is 86º; here we examine how much the critical supersaturation

is affected by changing the contact angle by up to plus/minus 10 degrees. As noted in the main text, when the contact angle

is given a new value, the model needs to be re-fitted to the adsorption data by changing the value of the FHH parameter $A$.

Figure A1 shows the values of $A$ as a function of the contact angle in the range 76º - 96º when this is done (the values of the

parameters $B$ and $s$ were kept at 1.93 and 75 nm, respectively). Figure A2 shows the experimental adsorption data together





with the model for $(\theta, A) = (76°, 7.9)$, $(86°, 12)$ and $(96°, 18)$. As can be seen, all curves fit the data well, although there is some deviation at low saturation ratios. Figure S3 shows the calculated critical supersaturations for water nucleation. The dry particle diameter (100 nm) was chosen based on the experimental nucleation data of Kotzick et al. (1997). The relative uncertainty of the critical supersaturation of water due to an uncertainty of $\pm5°$ in $\theta$ is quite insignificant, about 1.5%.

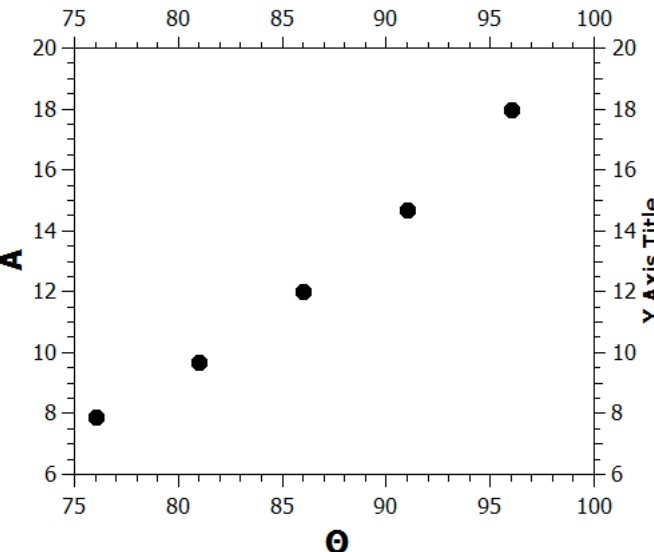


**Figure A1.** Pairs of contact angle and the FHH parameter $A$ that produce good model fits to the adsorption data.

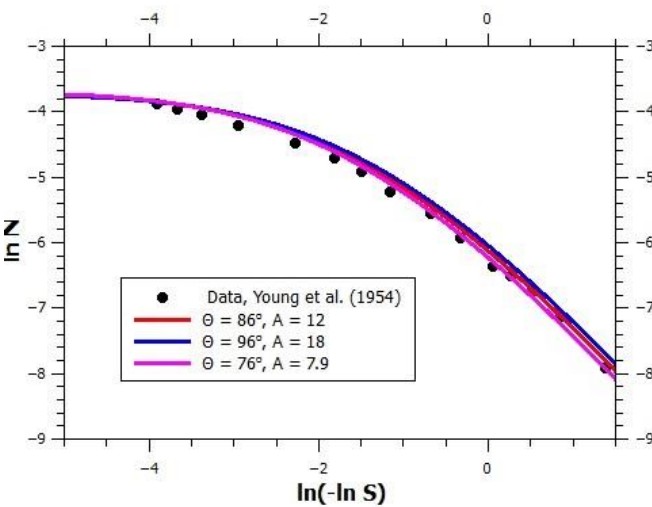

**Figure A2.** Model fits to adsorption data for three pairs of $\theta$ and $A$.




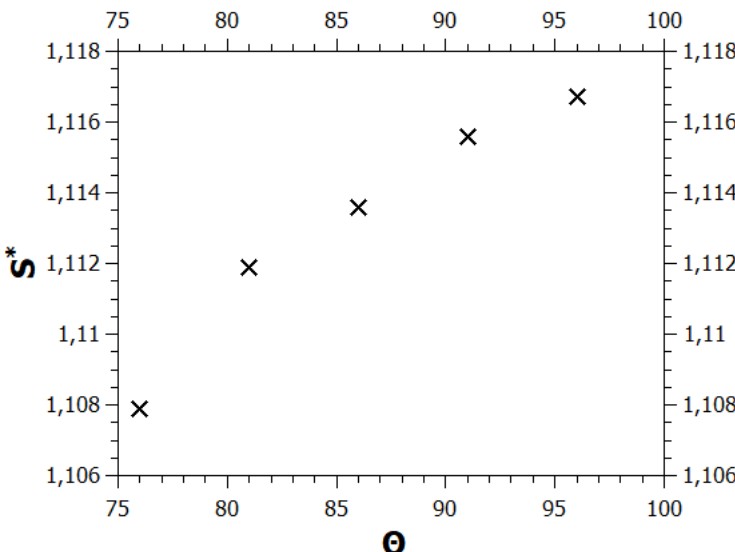

**Figure A3.** Critical supersaturations in water nucleation as a function of contact angle. The *A*-parameter values used in the
calculations are as shown in Fig. S1. Nucleation temperature is +25 ºC and dry particle diameter 100 nm.

*Acknowledgments*. This work was supported by the Academy of Finland, C-Main project (grant no. 309141) and the Center

of Excellence programme (grant no. 307331), and by the European Research Council (ERC) under the European Union's

Horizon 2020 research and innovation programme (grant agreement No. 717022).

*Code and Data availability*. Code and model results are available upon request

*Competing interests.* The authors declare that they have no conflict of interest.

*Author contributions*. AL performed all theoretical calculations and drafted the manuscript. All co-authors discussed the

results. JM and AN commented on the manuscript and contributed to writing of the final version.

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
