# Peer review of "Heterogeneous nucleation of water vapor on different types of black carbon particles"

_Atmospheric Chemistry and Physics, 2020_

## Short Comment (SC1) · 27 Apr 2020

To this very interesting work, I would like to suggest that the authors briefly address the role of contact region between soot spherules (also known as monomers) in their discussion.

Since real soot particles typically contain at least a few spherules, menisci between these spherules may alter the curvature of any condensed phase water. While this well-known phenomenon (Butt and Kappl, 2009) is probably outside of the scope of the present work, it may be worthwhile to note at what point and by approximately what degree it might alter the authors' conclusions. For example, since capillary menisci form below saturation, how would the authors conclusions change for a particle in

which capillary condensation is normally imagined?

I do realize that the macroscopic phenomenon of capillary condensation is not treated by a nucleation theory. I also realize that experimental data have already been presented. Hence, I expect that only a brief comment would be necessary to clarify this question. I apologize if such a comment was already made, and I missed it.

One might also hypothesize that the junctions between soot spherules may be important in the sense of being either more or less heterogeneous than the spherules. However, I am not aware of experimental evidence in support of this hypothesis.

Reference:

Butt, Hans Jürgen, and Michael Kappl. 2009. "Normal Capillary Forces." Advances in Colloid and Interface Science 146 (1–2): 48–60. https://doi.org/10.1016/j.cis.2008.10.002.

---

## Referee Comment (RC1) · Anonymous Referee #2 · 12 May 2020

The authors describe a new approach to parameterize the CCN activity of insoluble particles. Instead of the solute effect, they use the FHH-adsorption-isotherm to describe the water adsorption on the surface of insoluble carbon black particles. The manuscript is well written and well structured. The scientific content is exciting and fills a long-existing knowledge gap. Therefore, I only have very few minor comments.

P3 L53 "as well"

P3 L 58 "climate active" is a bit vague. I would suggest referring to radiative forcing and cloud formation directly

P7L130 It is a bit confusing when you say "droplet". Does this refer to a single droplet or to the water patches on the material

[Figure]

P11 L204 (Aria et al., 2016) investigated the wettability of graphene after exposure to ambient air. While this can be presentative for certain aspects of atmospheric black carbon particles, it should not be taken as an example for an atmospheric aging process. To point out, that exposure to atmospheric condition increases the interaction of black carbon with water, the following studies should be considered as an additional reference

Tritscher, T.; Jurányi, Z.; Martin, M.; Chirico, R.; Gysel, M.; Heringa, M. F.; De-Carlo, P. F.; Sierau, B.; Prévôt, A. S. H.; Weingartner, E.; et al. Changes of Hygroscopicity and Morphology during Ageing of Diesel Soot. Environ. Res. Lett. 2011, 6 (3), 34026. https://doi.org/10.1088/1748-9326/6/3/034026. Grimonprez, S.; Faccinetto, A.; Batut, S.; Wu, J.; Desgroux, P.; Petitprez, D. Cloud Condensation Nuclei from the Activation with Ozone of Soot Particles Sampled from a Kerosene Diffusion Flame. Aerosol Sci. Technol. 2018, 52 (8), 814–827. https://doi.org/10.1080/02786826.2018.1472367. Friebel, F.; Mensah, A. A. Ozone Concentration versus Temperature : Atmospheric Aging of Soot Particles. Langmuir 2019, 35 (45), 14437−14450. https://doi.org/10.1021/acs.langmuir.9b02372.

---

## Referee Comment (RC2) · Anonymous Referee #3 · 12 Aug 2020

The authors present measurements of heterogeneous nucleation of water on different types of carbon black. They tend to understand these measurements with a recently developed adsorption nucleation theory. The interesting finding is that with sufficient adsorption sites, soot particles can activate even when no soluble material is present, which is in contrast to the kappa-Köhler theory. The paper is well-written and fits into the journal ACP. However, the paper is short and eventually should be published as the new MS type "ACP Letters". Anyway, there are some shortcomings which should be discussed before publication:

Main comments

The contact angle is a rather imprecise value as mentioned by the authors (line 121-130). However, the explanation why in the case of graphite the contact angle should

be more trustable (+/-5°) is not very convincing. More arguments should be presented considering the structure, morphology and chemistry of the respective graphite type and its impact on the contact angle and on the heterogeneous nucleation.

Häusler et al. (2018) have measured the impact of graphene and graphene oxides on heterogeneous ice nucleation and found that structure, morphology and chemistry have an important impact. The authors might discuss how they can parameterize such findings for the heterogeneous nucleation of water vapor.

Niedermeier et al. (2014) have developed a soccer ball model for heterogeneous ice nucleation relying on different contact angles on the surface of a nucleus. The authors might discuss how this model compares to their adsorption nucleation theory.

Minor comments

Fig. 3: The caption of fig. 3 mentions more details in the text. However, the text does not explain how to read this figure and for what reason lnN is plotted against ln(-lnS). Fig. A1: The x-axis theta should have the unit degree (°). The label of the y-axis on the right-hand site should be deleted.

Fig. A3: The supersaturation S* should have the unit percent (%). The contact angle (theta) should have the unit degree (°)

References

Haeusler, T., Gebhardt, P., Iglesias, D., Rameshan, C., Marchesan, S., Eder, D., Grothe, H. (2018), Ice Nucleation Activity of Graphene and Graphene Oxides, J. Phys. Chem. C 122, 15, 8182–8190, https://doi.org/10.1021/acs.jpcc.7b10675

Niedermeier, D., B. Ervens, T. Clauss, J. Voigtländer, H. Wex, S. Hartmann, and F. Stratmann (2014), A computationally efficient description of heterogeneous freezing: A simplified version of the Soccer ball model, Geophys. Res. Lett., 41, 736–741, doi:10.1002/2013GL058684.

---

## Author Comment (AC1) · 10 Sep 2020

We thank all reviewers for useful comments. We would like to note that we found a minor bug in the program used for ANT calculations. After fixing the bug, the adsorption site distances used for calculating the coagulation nucleation curves in Fig. 4 are 4.2 and 4.4 nm instead of 3.6 and 3.8 nm. This change does not affect the discussion and conclusions in any way.

**Reviewer 2**

*The authors describe a new approach to parameterize the CCN activity of insoluble particles. Instead of the solute effect, they use the FHH-adsorption-isotherm to de-scribe the water adsorption on the surface of insoluble carbon black particles. The manuscript is well written and well structured. The scientific content is exciting and fills a long-existing knowledge gap. Therefore, I only have very few minor comments.*

We thank Reviewer 2, our replies to the detailed comments are below

*P3 L53 "as well"*
Corrected.

*P3 L 58 "climate active" is a bit vague. I would suggest referring to radiative forcing and cloud formation directly*

New formulation: … different types of black carbon (BC) particles, which have long been known to be climate active – but with large uncertainty on their impact direct radiative forcing and on cloud formation and lifetime.

*P7L130 It is a bit confusing when you say "droplet". Does this refer to a single droplet or to the water patches on the material*

It refers to a single droplet growing on a BC particle. We have changed the sentences as follows: A single droplet on a BC particle grows along the red curve 1 until it reaches the critical supersaturation $S_{cl}{}^*$ and nucleates. However, if there are several similar droplets growing on the particle, and the average distance between them is small enough so that they coalesce to form a uniform liquid film at sufficiently low saturation ratio…

*P11 L204 (Aria et al., 2016) investigated the wettability of graphene after exposure to ambient air. While this can be presentative for certain aspects of atmospheric black carbon particles, it should not be taken as an example for an atmospheric aging process. To point out, that exposure to atmospheric condition increases the interaction of black carbon with water, the following studies should be considered as an additional reference*

*Tritscher, T.; Jurányi, Z.; Martin, M.; Chirico, R.; Gysel, M.; Heringa, M. F.; De-Carlo, P. F.; Sierau, B.; Prévôt, A. S. H.; Weingartner, E.; et al.Changes of Hygroscopicity and Morphology during Ageing of Diesel Soot.Environ.Res.Lett. 2011, 6 (3), 34026. https://doi.org/10.1088/1748-9326/6/3/034026. Grimon-prez, S.; Faccinetto, A.; Batut, S.;*

*Wu, J.; Desgroux, P.; Petitprez, D. Cloud Condensation Nuclei from the Activation with Ozone of Soot Particles Sampled from a Kerosene Diffusion Flame.Aerosol Sci.Technol.2018, 52 (8), 814-827. https://doi.org/10.1080/02786826.2018.1472367. Friebel, F.; Mensah, A. A. Ozone Concentration versus Temperatureâ̈Ä́r: Atmospheric Aging of Soot Particles. Langmuir 2019, 35 (45), 14437–14450. https://doi.org/10.1021/acs.langmuir.9b02372*

We have added these references.

---

## Author Comment (AC2) · 10 Sep 2020

We thank all reviewers for useful comments. We would like to note that we found a minor bug in the program used for ANT calculations. After fixing the bug, the adsorption site distances used for calculating the coagulation nucleation curves in Fig. 4 are 4.2 and 4.4 nm instead of 3.6 and 3.8 nm. This change does not affect the discussion and conclusions in any way.

**Reviewer 3**

*The authors present measurements of heterogeneous nucleation of water on different types of carbon black. They tend to understand these measurements with a recently developed adsorption nucleation theory. The interesting finding is that with sufficient adsorption sites, soot particles can activate even when no soluble material is present, which is in contrast to the kappa-Köhler theory. The paper is well-written and fits into the journal ACP. However, the paper is short and eventually should be published as the new MS type "ACP Letters". Anyway, there are some shortcomings which should be discussed before publication:*

> We thank reviewer 3 and would like to note that although ms is not very long, it is clearly over the 2500 word limit set for ACP letters. Our detailed replies are below.

*Main comments The contact angle is a rather imprecise value as mentioned by the authors (line 121-130). However, the explanation why in the case of graphite the contact angle should be more trustable (+/-5∘) is not very convincing.*

> As a matter of fact, our sensitivity analysis shows that changing the contact angle by +/- 10 degrees and the FHH A-parameter simultaneously so that the FHH model still fits measured adsorption data well, the critical supersaturation is affected by less than 1% (we changed the y-axis of Fig. A3 to match those of Figs. 4 and 5). This is much less than the experimental uncertainty of the critical supersaturations of ozonolyzed Palas soot particles (Fig. 4). We have clarified this in the text, and removed to references to +/- 5 degrees.

*More arguments should be presented considering the structure, morphology and chemistry of the respective graphite type and its impact on the contact angle and on the heterogeneous nucleation. Häusler et al. (2018) have measured the impact of graphene and graphene oxides on heterogeneous ice nucleation and found that structure, morphology and chemistry have an important impact. The authors might discuss how they can parameterize such findings for the heterogeneous nucleation of water vapor.*

> Häusler et al. showed the importance of graphene lattice order as well as degree of surface oxidation and surface defects on immersion freezing. Compared with vapor-liquid nucleation, freezing is impacted by possible ice epitaxy, and therefore understanding the exact surface structure is even more important than in droplet formation. Nevertheless, we now discuss structure of the Palas soot in conjunction with possible impact of capillary menisci formed between primary soot spherules to the heterogeneous nucleation (see also our reply to J.C. Corbin).

*Niedermeier et al. (2014) have developed a soccer ball model for heterogeneous ice nucleation relying on different contact angles on the surface of a nucleus. The authors might discuss how this model compares to their adsorption nucleation theory.*

The soccer ball model is intermediate between stochastic and singular (or deterministic) nucleation theories. Classical nucleation theory is based on the idea that nucleation is stochastic phenomenon whereas ice nucleation theories that assume specific temperature dependent ice nucleation sites are deterministic. The adsorption nucleation theory is deterministic as one of its premises is that nucleation occurs on specific adsorption sites, and there is no stochastic element included in the theory. Such an element could be introduced similarly as in the soccer ball model, by assuming a distribution of contact angles between the adsorption sites. However, without knowledge of e.g. the possible widths of such distributions, we prefer not to include this idea at present. Having said that, molecular dynamics could possibly be used in the future to explore this issue with chemically and physically heterogeneous surfaces. We have added these considerations in the manuscript.

*Minor comments*
*Fig. 3: The caption of fig. 3 mentions more details in the text. However, the text doesnot explain how to read this figure and for what reason lnN is plotted against ln(-lnS).*

The reason is that according to the FHH theory, multilayer adsorption data should align linearly in such a plot. This has been added to the text.

*Fig. A1: The x-axis theta should have the unit degree ($\circ$). The label of the y-axis onthe right-hand site should be deleted.Fig.*

Fixed

*A3: The supersaturation S\* should have the unit percent (%). The contact angle(theta) should have the unit degree ($\circ$)*

Fixed

*References Haeusler, T., Gebhardt, P., Iglesias, D., Rameshan, C., Marchesan, S., Eder, D.,Grothe, H. (2018), Ice Nucleation Activity of Graphene and Graphene Oxides, J. Phys.Chem. C 122, 15, 8182–8190, https://doi.org/10.1021/acs.jpcc.7b10675Niedermeier, D., B. Ervens, T. Clauss,J. Voigtländer, H. Wex, S. Hartmann,and F.Stratmann (2014), A computationally efficient description of heterogeneous freezing:A simplified version of the Soccer ball model, Geophys. Res. Lett.,41, 736–741,doi:10.1002/2013GL058684.*

---

## Author Comment (AC3) · 10 Sep 2020

We thank all reviewers for useful comments. We would like to note that we found a minor bug in the program used for ANT calculations. After fixing the bug, the adsorption site distances used for calculating the coagulation nucleation curves in Fig. 4 are 4.2 and 4.4 nm instead of 3.6 and 3.8 nm. This change does not affect the discussion and conclusions in any way.

**J. C. Corbin**

*To this very interesting work, I would like to suggest that the authors briefly address the role of contact region between soot spherules (also known as monomers) in their discussion. Since real soot particles typically contain at least a few spherules, menisci between these spherules may alter the curvature of any condensed phase water. While this well-known phenomenon (Butt and Kappl, 2009) is probably outside of the scope of the present work, it may be worthwhile to note at what point and by approximately what degree it might alter the authors' conclusions. For example, since capillary menisci form below saturation, how would the authors conclusions change for a particle in*
*which capillary condensation is normally imagined? I do realize that the macroscopic phenomenon of capillary condensation is not treated by a nucleation theory. I also realize that experimental data have already been presented. Hence, I expect that only a brief comment would be necessary to clarify this question. I apologize if such a comment was already made, and I missed it. One might also hypothesize that the junctions between soot spherules may be important in the sense of being either more or less heterogeneous than the spherules. However, I am not aware of experimental evidence in support of this hypothesis.*
*Reference:Butt,Hans Jürgen,and Michael Kappl.2009."Normal CapillaryForces."Advances in Colloid and Interface Science 146 (1–2):48–60.https://doi.org/10.1016/j.cis.2008.10.002.*

We would like to thank Dr. Corbin for this useful insight. There are two factors impacting possible capillary condensation. The first is that the contact angle we use is nearly 90 degrees, which means that a meniscus cannot in practice be concave, which in turn implies that capillary condensation will not occur at undersaturation. The second is the primary spherule sizes. Palas soot primary spherules have diameters on the order of 5-10 nm. Crouzet and Marlow (Aerosol Sci. Tech. 22, 43-59, 1995) have modelled the nucleation of pendular water ring formed around the contact point of two equal sized spheres. It can be read from their Fig. 6 and Table 2 that when contact angle is 85 degrees, the supersaturation required for nucleation to occur is about 8%. However, this is when the sphere radius is 50 nm. It can also be seen from Table 2 that the critical supersaturation increases rapidly with sphere radius; for 200 nm spheres it would be just 2%. We can thus conclude that with Palas soot, it is unlikely that this type of capillary nucleation would take place at significantly lower supersaturation than we have calculated for our Fig. 4. We have added discussion in the manuscript.